# Yes-Associated Protein 1 Is a Novel Calcium Sensing Receptor Target in Human Parathyroid Tumors

**DOI:** 10.3390/ijms22042016

**Published:** 2021-02-18

**Authors:** Giulia Stefania Tavanti, Chiara Verdelli, Annamaria Morotti, Paola Maroni, Vito Guarnieri, Alfredo Scillitani, Rosamaria Silipigni, Silvana Guerneri, Riccardo Maggiore, Gilberto Mari, Leonardo Vicentini, Paolo Dalino Ciaramella, Valentina Vaira, Sabrina Corbetta

**Affiliations:** 1Laboratory of Experimental Endocrinology, IRCCS Istituto Ortopedico Galeazzi, 20161 Milan, Italy; giulia.tavanti@unimi.it (G.S.T.); chiara.verdelli@grupposandonato.it (C.V.); 2Department of Biomedical, Surgical and Dental Sciences, University of Milan, 20122 Milan, Italy; 3Division of Pathology, Fondazione IRCCS Ca’ Granda, Ospedale Maggiore Policlinico, 20122 Milan, Italy; annamaria.morotti@policlinico.mi.it (A.M.); valentina.vaira@unimi.it (V.V.); 4Laboratory of Experimental Biochemistry & Molecular Biology, IRCCS Istituto Ortopedico Galeazzi, 20161 Milan, Italy; paola.maroni@grupposandonato.it; 5Division of Medical Genetics, Fondazione IRCCS Ospedale Casa Sollievo della Sofferenza, 71013 San Giovanni Rotondo, Italy; vitoguarnieri@gmail.com; 6Endocrinology Unit, IRCCS Ospedale Casa Sollievo della Sofferenza, 71013 San Giovanni Rotondo, Italy; a.scillitani@operapadrepio.it; 7Medical Genetics Laboratory, Fondazione IRCCS Ca’ Granda, Ospedale Maggiore Policlinico, 20122 Milan, Italy; rosamaria.silipigni@policlinico.mi.it (R.S.); s.guerneri@policlinico.mi.it (S.G.); 8Endocrine Surgery, IRCCS Ospedale San Raffaele, 20123 Milan, Italy; maggiore.riccardo@hsr.it (R.M.); mari.gilberto@hsr.it (G.M.); 9Endocrine Surgery, IRCCS Istituto Auxologico, 20122 Milan, Italy; viceleo@hotmail.com; 10Endocrinology Unit, ASST Grande Ospedale Metropolitano Niguarda, 20162 Milan, Italy; pdalino@icloud.com; 11Department of Pathophysiology and Organ Transplantation, University of Milan, 20122 Milan, Italy; 12Endocrinology and Diabetology Service, IRCCS Istituto Ortopedico Galeazzi, 20161 Milan, Italy

**Keywords:** YAP1, LATS1/2, CASR, parathormone, MEN1, parathyroid tumors

## Abstract

The Hippo pathway is involved in human tumorigenesis and tissue repair. Here, we investigated the Hippo coactivator Yes-associated protein 1 (YAP1) and the kinase large tumor suppressor 1/2 (LATS1/2) in tumors of the parathyroid glands, which are almost invariably associated with primary hyperparathyroidism. Compared with normal parathyroid glands, parathyroid adenomas (PAds) and carcinomas show variably but reduced nuclear YAP1 expression. The kinase LATS1/2, which phosphorylates YAP1 thus promoting its degradation, was also variably reduced in PAds. Further, *YAP1* silencing reduces the expression of the key parathyroid oncosuppressor *multiple endocrine neoplasia type 1*
*(MEN1)*, while *MEN1* silencing increases *YAP1* expression. Treatment of patient-derived PAds-primary cell cultures and Human embryonic kidney 293A (HEK293A) cells expressing the calcium-sensing receptor (CASR) with the CASR agonist R568 induces YAP1 nuclear accumulation. This effect was prevented by the incubation of the cells with RhoA/Rho-associated coiled-coil-containing protein kinase (ROCK) inhibitors Y27632 and H1152. Lastly, CASR activation increased the expression of the YAP1 gene targets *CYR61*, *CTGF*, and *WNT5A*, and this effect was blunted by *YAP1* silencing. Concluding, here we provide preliminary evidence of the involvement of the Hippo pathway in human tumor parathyroid cells and of the existence of a CASR-ROCK-YAP1 axis. We propose a tumor suppressor role for YAP1 and LATS1/2 in parathyroid tumors.

## 1. Introduction

The Hippo pathway is a central cellular pathway, which regulates homeostasis and plays prominent roles in carcinogenesis and regenerative processes. YAP1 (Yes-associated protein 1) and its paralog transcriptional coactivator with PDZ-binding motif, Tafazzin (TAZ), which are the principal effectors of the Hippo signaling pathway, are relatively inactive in adults to maintain organ homeostasis, while they are robustly induced during neoplastic disorders [1,2,3]. The Hippo-YAP/TAZ pathway has emerged as a hub that integrates diverse stimuli, including mechanical and cytoskeletal cues, cell adhesion, apico-basolateral polarity, and mitogens, to control cell growth and organ size [4]. The Hippo tumor suppressor pathway functions to inhibit the activity of YAP/TAZ transcriptional coactivators. The Hippo pathway is complex and involves the MST1 and MST2 (Mammalian STE20-like protein kinase 1/2), LATS1 and LATS2 (large tumor suppressors 1 and 2), and adaptor proteins SAV1 (Salvador homolog 1) and MOB kinase activator 1A and 1B (MOB1A and MOB1B). MST1 and MST2 facilitate LATS1 and LATS2 phosphorylation, which in turn facilitate LATS-dependent phosphorylation of YAP and TAZ [5]. Genetic and epigenetic alterations of the tumor suppressor LATS1/2 have been observed in several cancers [6]. Phosphorylation of the complex YAP/TAZ results in degradation through ubiquitination or cytoplasmic retention via 14-3-3 binding; moreover, it further promotes β-TrCP-mediated YAP/TAZ ubiquitination and degradation. Therefore, upon the inhibition of the Hippo pathway, YAP/TAZ are activated and translocate into the nucleus to bind the TEAD (transcriptional enhanced associate domain) family of transcription factors, thus promoting cell proliferation, stem cell self-renewal, and tumorigenesis [7].

Parathyroid tumors, which are mainly benign, are the second most frequent endocrine neoplasia clinically associated with the endocrine disorder known as primary hyperparathyroidism, which represents a frequently occurring cause of osteoporosis and bone fragility. Primary hyperparathyroidism is characterized by hypercalcemia and concomitant inappropriately elevated parathormone (PTH) release from tumors of the parathyroid glands. 

The calcium-sensing receptor (CASR) is a key molecule involved in the exquisite parathyroid function of sensing extracellular calcium concentrations ([Ca^2+^]_o_) and consequently of modulating PTH release and synthesis. CASR is a pleiotropic, type III G protein-coupled receptor (GPCR) that functionally associates with the cytoskeletal protein filamin A [8,9]. The CASR has been shown to activate a wide range of intracellular pathways through coupling to different Gα proteins in a tissue- and cell-type-specific manner [8]. CASR is mainly coupled to Gα protein q/11 in parathyroid cells [10], through which the extracellular signal-regulated kinase (ERK) signaling activates [8,11]. CASR has also been reported to activate the small GTPase protein RhoA, presumably through Gα 12/13 and/or Gαq/11 [12,13,14].

The Hippo pathway has never been investigated in parathyroid tumors until now, while some evidence suggests its potential role in parathyroid tumorigenesis: (1) *YAP1* gene maps on chromosome 11 in 11q22.1, a region frequently affected by the loss of heterozygosity in parathyroid tumors [15,16]; (2) recent experimental data identified *LATS2* gene as a target of the aberrantly expressed miR-372, which is overexpressed in a subset of parathyroid tumors [17]; (3) CASR, a crucial molecule in parathyroid tumors, might be coupled to Hippo signaling through RhoA/Rho-associated protein kinase (ROCK) activation [12]. Recent studies uncovered the critical role of G protein-coupled receptors (GPCR) signaling in YAP/TAZ regulation [18,19,20,21]. ROCK is identified as a critical downstream effector of GTPase RhoA, which contains two isoforms, ROCK1 and ROCK2. ROCKs have a principal function in the generation of actin–myosin contractility and regulation of actin cytoskeleton dynamics. Moreover, they regulate various cellular functions, such as apoptosis, growth, migration, metabolism, and cellular contraction [22].

Here, we analyzed the expression and function of the Hippo pathway master regulator YAP1 in human parathyroid tissues and its interconnection with the parathyroid key genes *multiple endocrine neoplasia type 1 (MEN1)* and *CASR*. 

## 2. Results

### 2.1. YA1P Expression in Human Parathyroid Tumors

The expression of the Hippo cofactor YAP1 was investigated by immunohistochemistry in formalin-fixed paraffin-embedded (FFPE) sections from normal parathyroid glands (PaNs, *n* = 4), parathyroid adenomas (PAds, *n* = 11), and parathyroid carcinomas (PCas, *n* = 6) (Figure 1). Normal parathyroid glands from normocalcemic patients showed that a consistent subset of the parathyroid cells expressed YAP1 in the nuclei, whereas, and consistent with its role of a transcription factor, cytoplasmic expression was relatively weak (Figure 1a,b). Of note, parathyroid cells of the rim of normal tissue surrounding adenomas showed intense nuclear YAP1 expression (Figure 1c, black arrow). Compared with normal samples, PAds showed variable but similar nuclear expression of YAP1 (Figure 1c,d,h). By contrast, parathyroid carcinomas (PCas) showed a remarkable loss of YAP1 nuclear staining (Figure 1e,f,h) irrespective of the *cell division cycle 73 (CDC73)* or *MEN1* status (Figure 1h). 

We then analyzed by immunoblotting the expression of LATS1/2, phosphorylated YAP1, and total YAP1 proteins in a series of PAds and human embryonic kidney 293A (HEK293A) cells. Due to the unavailability of fresh parathyroid normal glands for ethical reasons, we used a HEK293A cell model as a surrogate non-neoplastic control. In PAds, the expression of the YAP1 proteins was reduced compared with that detected in HEK293A cells (Figure 1i), while the proteins LATS1, LATS2, and phosphorylated LATS1 (pLATS1) were variably reduced among the samples. 

This set of data suggested that the Hippo pathway cofactor YAP1 may act as an oncosuppressor in parathyroid tumorigenesis rather than as an oncogene. 

### 2.2. MEN1 Aberrations Do Not Directly Modulate YAP1 Expression in Parathyroid Tumors 

The *YAP1* gene maps on chromosome 11q22.1, a region frequently interested in the loss of heterozygosity (chr.11 LOH) in parathyroid tumors [17,18]. Therefore, we tested the hypothesis that the YAP1 reduction detected in parathyroid tumors could be caused by chr.11 LOH. PAds cytogenetic analysis showed that LOH at chr.11 occurred in 8 out of the 20 analyzed cases (40%). *YAP1* expression levels were similar in chr.11 LOH-PAds and chr.11 WT-PAds (Figure 2a). However, *YAP1* transcripts positively correlated with *MEN1* expression levels in adenomas with chr.11 LOH (Figure 2b), but not in tumors with a normal chr.11 haplotype (Figure 2c). Of note, transient silencing of *MEN1* in PAds-derived cells (Figure 2d) induced a small but significant increase in *YAP1* expression (Figure 2e). On the other hand, *YAP1* silencing in HEK293A cells (Figure 2f) significantly reduced *MEN1* expression (Figure 2g). These findings suggest that the two genes mapping on chr.11 may interact by modulating their expression in tumor parathyroid cells.

### 2.3. CASR Activation Induces Nuclear YAP1 Accumulation and Inhibits LATS2 Expression

The YAP1 expression pattern detected in parathyroid tumors reflects the heterogeneous expression of the CASR protein, which has been extensively reported in previous studies [23,24]. In the series of parathyroid adenomas, coexpression of YAP1 and CASR was analyzed by immunohistochemistry (IHC) (Figure 3). In line with our hypothesis, parathyroid tumor cells with an intense CASR staining showed YAP1 nuclear expression.

Therefore, we tested the hypothesis that YAP1 nuclear accumulation could be involved in the CASR-mediated extracellular calcium sensing in human parathyroid cells. We forced CASR activation by incubating primary PAd cultures with increasing concentrations of the CASR agonist R568. Results showed that CASR activation induced nuclear YAP1 accumulation (Figure 4a). Since the cytoplasmic pool of YAP1 was significantly reduced by CASR activation (Figure 4b), we could surmise that CASR might modulate the availability of YAP1 as a transcription factor in the nuclei of parathyroid adenomatous cells. Moreover, CASR activation by R568 modestly affected LATS1, pLATS1, and LATS2 proteins, reducing their abundance (Figure 4c).

### 2.4. CASR Activation Modulates the Expression of YAP1 Target Genes in PAds-Derived Cells

CASR activation by R568 (0.5 and 5.0 μM) induced the expression of the YAP1 target genes *CYR61*, *CTGF*, and *WNT5A* in PAds-derived primary cultures (Figure 4d–f). When we transiently silenced *YAP1* in primary PAds-cultures, the CASR agonist R568 ceased to induce *CYR61*, *CTGF*, and *WNT5A* mRNA levels (Figure 4g). On the other hand, *PTH* transcript expression was unaffected by R568 stimulation as well as by *YAP1* silencing. Moreover, R568 induced increases in *GCM2* mRNA levels, though they were unaltered by *YAP1* silencing. *TBX1* and *CASR* transcripts were unaffected by R568 stimulation as well as by *YAP1* silencing, while *VDR* mRNA levels were reduced by R568 stimulation, though *YAP1* silencing did not exert any significant effect (Figure 4h). 

### 2.5. Dissecting CASR-YAP1 Signaling in CASR-HEK293A Cells 

To gain insights into CASR-YAP1 signaling, we used the CASR-HEK293A cell model (Appendix A), which showed efficient CASR activity as measured by an increase in phosphorylated ERK after stimulation with [Ca^2+^]_o_ and R568 (Appendix A). In CASR-HEK293A cells, R568 induced YAP1 accumulation in the nuclei (Figure 5a), while changes in the cytoplasmic fraction were marginal (Figure 5b). No effect was detected in mock-transfected HEK293A cells treated with R568 (Appendix A). Further, R568 incubation also inhibited phosphorylated and total LATS1 and LATS2 expression (Figure 5c). 

The reduction in phosphorylated cytoplasmic YAP1 observed after CASR activation was abolished by incubation with both the RhoA inhibitors Y27632 or H1152 (Figure 6a–c), although total cytoplasmic YAP1 levels were unchanged (Figure 6c–e). Lastly, nuclear YAP1 accumulation induced by R568 stimulation was abolished upon treatment with the two RhoA inhibitors (Figure 6f–h).

In keeping with our hypothesis about a CASR-YAP1 axis in parathyroid tumorigenesis, CASR activation by R568 upregulated *CYR61*, *CTGF*, and *MYC* expression in CASR-HEK293A cells. This effect was abolished by *YAP1* silencing (Figure 7a,b), while *WNT5A* expression, at variance with that observed in PAds-derived cells, was significantly reduced by CASR activation and further diminished by *YAP1* silencing (Figure 7c).

## 3. Discussion

Here, we first report that the Hippo signaling cofactor YAP1 and the upstream components LATS1/2 are expressed and deregulated in parathyroid tumors. Nuclear YAP1 accumulation was evident in a substantial proportion of parathyroid epithelial cells in normal parathyroid glands and parathyroid adenomas, while the accumulation of YAP1 in the cytosol was variably reduced in cells of parathyroid adenomas, which sometimes showed a more intense nuclear staining compared with cells in normal glands. In line with this observation and with the repressive nature of LATS1/2, these two proteins were variably reduced in parathyroid adenomas, suggesting a state of relative activation of the Hippo pathway in benign parathyroid tumors. The reduced LATS2 levels detected in parathyroid adenomas are in agreement with the inhibitory effect on *LATS2* expression levels exerted by miR-372, which has been found to be overexpressed in a subset of parathyroid adenomas and most parathyroid cancers [17]. 

By contrast, in parathyroid cancers, YAP1 was variably downregulated in the cytoplasm and at the nuclear level. In particular, parathyroid cancers harboring *CDC73* inactivating mutations, a frequent genetic aberration occurring in about 70% of parathyroid cancers [25,26,27], showed a reduced proportion of YAP1 nuclear positive cells, while they were lost in samples harboring the *CDC73* wild type allele. This pattern of expression suggests that *YAP1* may act as an oncosuppressor in parathyroid tumorigenesis, at variance with reports in most common human cancers [3]. Though YAP1 is assumed to act as an oncoprotein, an increasing number of studies have reported that YAP1 may have tumor-suppressor and pro-apoptotic functions, suggesting a paradoxical role of YAP1, probably dependent on the cellular context and binding partners [7,28,29]. For example, YAP can bind to DNA-binding tumor suppressors, including RUNXs and p73, which induce cell cycle arrest and apoptosis [30,31]. Similarly, in multiple myeloma, YAP is deleted or consistently downregulated to evade apoptosis despite pervasive DNA damage [32]. Of note, parafibromin, encoded by the *CDC73* gene, may be involved in YAP1 modulation. Physical interaction between YAP/TAZ and parafibromin has been demonstrated through the formation of a complex, which inversely regulates the activities of YAP and TAZ depending on the parafibromin phosphorylation status; in particular, the coactivator function of YAP on TEAD transcription factor was specifically potentiated by tyrosine-phosphorylated parafibromin [33]. Therefore, we are tempted to speculate that the loss of parafibromin in parathyroid cancer cells [26] contributes to the reduction in YAP1 nuclear accumulation and transcriptional function. Nonetheless, the loss of YAP1 nuclear staining in cells harboring the *CDC73* wildtype allele prompts further molecular mechanisms. 

Besides *CDC73*, *MEN1* is a master oncosuppressor involved in parathyroid tumorigenesis [34]. Chr.11 LOH can be detected in at least 40% of parathyroid tumors [15,16]. Though *YAP1* mRNA levels were similar in parathyroid adenomas harboring the chr.11 LOH and in those harboring the wildtype haplotype, they positively correlated with *MEN1* transcripts in parathyroid adenomas harboring chr.11-LOH. In line with this observation, *YAP1* silencing decreased *MEN1* transcripts in HEK293A cells, indicating that YAP1 deregulation may contribute to modulate *MEN1* expression aberrantly. Moreover, menin, the protein product encoded by *MEN1*, can modulate the expression of YAP1: In human hepatocellular carcinomas menin epigenetically upregulates *YAP1* gene expression [35]. By contrast, in primary PAds cultures, the loss of *MEN1* slightly increased *YAP1* expression. Therefore, *YAP1* and *MEN1* mutually and differently regulated their expression levels, suggesting that genetic and epigenetic [36] aberrations occurring in parathyroid tumors may determine the degree of activity of the two genes and their target genes. Certainly, our data did not provide final insights into the mechanisms underlying such interaction. Indeed, YAP1 nuclear accumulation may be alternatively the downstream result of other factors. 

The *YAP1* variable downregulation observed in parathyroid tumor tissues resembles the extensively described heterogeneous expression of CASR in these tumors [23,24,37,38]. CASR is a key molecule in parathyroid pathophysiology: It mediates the parathyroid cell sensitivity to extracellular calcium concentrations resulting in inhibition of the PTH release; it further acts as an oncosuppressor in parathyroid cells as *CASR* inactivating mutations induce parathyroid hyperplasia in the set of familial hypocalciuric hypercalcemia type 1 (FHH1; OMIM#145980). Here, we demonstrated that YAP1 is a downstream effector of CASR signaling in parathyroid cells (Figure 8). 

Activation of CASR by treatment with increasing concentrations of the CASR allosteric agonist R568 increased YAP1 in the nuclear fraction and reduced LATS1/2 in the cytoplasmic fraction in PAds-derived cells, suggesting that CASR activation is coupled to Hippo pathway inhibition and, consequently, YAP1 nuclear accumulation. This observation is in line with the consistent YAP1 expression observed in the nuclei of normal parathyroid cells and adenomatous parathyroid cells by immunohistochemistry. Indeed, we detected a great variability in the number of cells with YAP1 positive nuclear immunostaining, and these findings are likely related to the variable downregulation of the CASR in parathyroid tumors. Moreover, CASR activation is coupled to nuclear YAP1 accumulation through ROCK activation, in line with a previous report describing the coupling of CASR to Rho GTPase RhoA activation [12,13]. In parathyroid tumors, Gα_q/11_ and *FLNA*/filamin A are downregulated at mRNA and protein levels [9,10]. Filamin A binds to and promotes activation of RhoA [39]; therefore, filamin A and Gα_q/11_ downregulation occurring in parathyroid tumors may inhibit RhoA activation and the inhibitory effect of the Hippo signaling pathway on YAP1 [40], contributing to reduction in YAP1 nuclear accumulation. Therefore, in parathyroid adenomas, downregulated CASR-related proteins are associated with low YAP1 nuclear accumulation and, likely, activity. 

YAP1 nuclear accumulation is associated with YAP1 transcriptional activity in PAds-derived cells. CASR activation increased the expression of the *CYR61*, *CTGF*, and *WNT5A* genes, which are targets of the YAP/TAZ/TEAD interaction in PAds-derived cultures and CASR-HEK293A cells. This effect was specifically mediated by YAP1 activation as YAP1 silencing blunted the R568-induced changes in *CYR61*, *CTGF*, and *WNT5A* expression. *CYR61/CCN1* and *CTGF/CCN2* were originally identified as early-response genes related to cell growth; CTGF/CCN2 is also known to be involved in the mitogen-activated protein kinase/extracellular signal-regulated kinase (MAPK/ERK) signaling pathway, which participates in the modulation of parathormone release from parathyroid cells [11]. CYR61/CCN1 and CTGF/CCN2 are variably deregulated in human cancers [41]. WNT5A is a target of YAP/TAZ/TEAD activation, and increased WNT5A may both further stimulate LATS1/2-YAP signaling and inhibit the WNT3A-stimulated β-catenin pathway [42]. Therefore, receptor and post-receptor downregulation of the CASR signaling may be associated with downregulated *WNT5A*, in line with the previously reported *WNT5A* downregulation in parathyroid tumors [43]. In addition, *YAP1* silencing did not affect *PTH, CASR*, *TBX1*, and *VDR* expression in PAds-derived cells, while *YAP1* signaling may be partially involved in R568-stimulated *GCM2* expression levels. GCM2 is a parathyroid-specific transcription factor involved in parathyroid embryogenesis; during adult life, GCM2 maintains CASR expression [44], and downregulated *GCM2* transcripts have been reported in parathyroid adenomas [45]. 

## 4. Materials and Methods 

### 4.1. Parathyroid Tissue Samples

Fresh samples from 20 PAds surgically removed from patients affected with primary hyperparathyroidism were collected and used for gene expression analysis in total tissue RNAs; biochemical, hormonal, and clinical characteristics have been previously published [46]. Fresh samples from further 10 PAds were collected immediately after surgical removal, partly snap-frozen, and partly dissociated for cell cultures. In all patients, fasting total serum calcium, phosphate, creatinine, and PTH were routinely measured to diagnose primary hyperparathyroidism.

This study was approved by the Institutional Ethical Committee (Ospedale San Raffaele Ethical Committee, protocol no. GPRC6A PARA, 07/03/2019; CE40/2019), and informed consent was obtained from all patients.

### 4.2. Cell Cultures

Samples from 10 PAds were cut into 1 mm^3^ fragments, washed with PBS, and partially digested with 2 mg/mL collagenase type I (Worthington, Lakewood, NJ, USA) for 90 min. Digested tissues were filtered with a cell strainer (100 μm Nylon, BD Falcon, Rignano Flaminio, Italy) to obtain a single-cell suspension and cultured in DMEM supplemented with 10% fetal bovine serum, 2 mmol/L glutamine, and 100 U/mL penicillin–streptomycin.

The human embryonic kidney HEK293A cell lines (INVITROGEN, CATALOG N.R705-07) were maintained until 30 passages and cultured in the same medium of PAds-derived cells, as described above. 

### 4.3. DNA Extraction and Array Comparative Genomic Hybridization (aCGH) Analysis

Genomic DNA from 20 PAds was isolated using Trizol reagent (Invitrogen). The array-CGH analysis was performed using 60-mer oligonucleotide probe technology (SurePrint G3 Human CGH 8 x 60K, Agilent Technologies, Santa Clara, CA, USA), according to the manufacturer instructions. The Feature Extraction and Cytogenomics 3.0.4.1, with the ADM-2 algorithm (Agilent Technologies, Santa Clara, CA, USA), were used for data analysis. To improve results accuracy, the Diploid Peak Centralization algorithm was also applied. To call for aberration, we set as threshold a minimum of five consecutive probes/regions and a Minimum Absolute Average Log Ratio (MAALR) of ±0.25. To identify lower levels of mosaicism, a second analysis was run with a MAALR of ±0.15. Only copy number variants not already reported in the public database of genomic variants (http://projects.tcga.ca/variation/, accessed on 1 January 2021) were listed. The GRch37/hg19 of the Human Genome Reference (March 2009) consortium was used as the reference genome. Data were partially previously published [46].

### 4.4. Peripheral Blood DNA Extraction and Direct Sequencing of CDC73 and MEN1 Genes

DNA was extracted from peripheral whole blood of patients affected with PCas with a classic phenol–chloroform protocol or from paraffin-embedded tumoral tissues. Coding sequences of the *CDC73* and *MEN1* genes were PCR amplified and directly sequenced as previously reported [25,47]. Gene aberrations were detected and reported in Table 1. 

### 4.5. RNA Isolation and Real-Time Quantitative Reverse Transcription (qRT-PCR)

Total RNA from PAds-derived cells and HEK293 cell cultures was extracted using TRIzol reagent (Invitrogen, ThermoFisher Scientific, Monza, MB, Italy) following the manufacturer’s recommendations. DNA contamination was removed by DNase I (Life Technologies, ThermoFisher Scientific, Monza, MB, Italy) treatment, and RNA was quantified by spectrophotometry at 260 nm.

Total DNA-free RNA (300 ng) was reverse-transcribed using the iScript cDNA Synthesis Kit (Bio-Rad Laboratories Inc., Segrate, MI, Italy). Then, cDNA was amplified using the TaqMan gene expression assay (probe numbers are listed in Table 2) and a StepOne Plus System. The reference genes HMBS and B2M were used to normalize expression data and obtain relative gene expression using the 2−DCt formula. All reagents and instruments were from ThermoFisher Scientific.

### 4.6. Protein Extraction and Western Blot Analysis

Cells and tissue samples were homogenized using NP40 lysis buffer (FNN0021, ThermoFisher) supplemented with protease and phosphatase inhibitors to obtain total protein extractions. Nuclear and cytoplasmic protein fractions were obtained using the Subcellular Protein Fractionation Kit for Cultured Cells (ThermoFisher Scientific). Protein concentration in each sample was determined by the bicinchoninic acid (BCA) method with the Pierce BCA Protein Assay Kit (ThermoFisher Scientific). Proteins were separated on 7% or 10% gels by SDS-PAGE, transferred onto a nitrocellulose membrane (Amersham Protran GE Healthcare Life Science), and probed with the following primary antibodies: CASR (ab19347, Abcam), phosphorylated YAP (Ser127) (#13008), YAP (#14074), phosphorylated LATS1 (Thr1079) (#8654), total LATS1 (#3477), and total LATS2 (#5888), all from Cell Signaling Technologies. GAPDH (ab9485) and Histone H3 (ab1791), both from Abcam, were used as loading controls for cytoplasmic and nuclear proteins, respectively. The binding of appropriate HRP-conjugated secondary antibodies was revealed using the chemiluminescence ChemiDoc XRS System (Bio-Rad). Analyses of bands densitometries were performed using Image Lab software (Bio-Rad), and protein expression levels were normalized using GAPDH or H3 as reference.

### 4.7. Immunohistochemistry

The samples were collected from 10 PAds, 6 PCas, and 4 PaNs incidentally removed from normocalcemic patients treated with thyroid surgery. Diagnosis of PCas was performed according to World Health Organization guidelines [50]. Sections of FFPE parathyroid tissues, after antigen retrieval, were incubated overnight at 4 °C with a rabbit monoclonal antibody specific for YAP1 (#14074, Cell Signaling Technologies). Immunostaining was performed with a streptavidin–biotin system (ABC kit, Santa-Cruz Biotechnology) and detected by diaminobenzidine (Novolink Polymer Detection System, Novocastra Laboratories, Leica Microsystems). Counterstaining was performed with Meyer’s hematoxylin solution. Negative-control sections were subjected to the same staining procedure without the primary antibody. Immunoreactivity was checked by light microscopy (CKX41 Olympus, Olympus Co., Tokyo, Japan). Double staining was performed on an automated system (Dako Omnis). Prediluted antibodies YAP1 and CASR were stained in brown with DAB and in red with Fast Red, respectively. Digital images were acquired with an Aperio slide scanner (Leica) and analyzed with Aperio ImageScope v.12.3.3. 

### 4.8. YAP1 and MEN1 Gene Silencing

For transient RNA interference experiments, HEK293A cells and PAds-derived primary cultures were seeded in 6-well plates at 1.5 × 10^5^ cells/well density. YAP1 silencing was performed using Dharmafect (T-2001-01; Dharmacon) as the transfection agent and YAP1 direct siRNA (L-012200-00-005; ON-TARGET Plus siRNA SmartPool Dharmacon) or control siRNA (D-001810-10-05, ON-TARGET Non-Targeting Plus). For *MEN1* silencing, cells were transiently transfected using Lipofectamine 3000 (Invitrogen–ThermoFisher Scientific) with either a *MEN1*-directed siRNA (EHU067451, Mission EsiRNA; Sigma–Aldrich, Merck Life Science S.r.l., Milan, Italy) or negative control siRNA (SIC001, Mission siRNA Universal negative control; Sigma–Aldrich). All the transfection experiments were conducted in Opti-MEM media (Gibco, ThermoFisher Scientific). Transfection mixtures were left for 5 h after which media was replaced with a standard one. After 48 h, transfected cells were used for further experiments and analyzed by qRT-PCR and western blot (WB).

### 4.9. CASR Transfection

HEK293A cells were cultured in antibiotic-free DMEM until they reached 70% confluence, and they were transiently transfected with a plasmid encoding for CASR, obtained by site-directed mutagenesis, as previously described [9]. Four micrograms CASR plasmid were transfected using TurboFect Transfection Reagent (R0533, ThermoFisher) in DMEM serum-free medium, according to the manufacturer’s instructions. Preliminary experiments were performed in order to determine the optimal concentrations of plasmid and to define the transfection settings. 

### 4.10. Treatments of CASR-HEK293A and PAds-Derived Cells with [Ca^2+^]_o_ or R568

Before treatments, the complete culture medium was replaced with a serum starvation medium (serum-free medium supplemented with 0.2% BSA and 1% l-Glutamine), and cells were incubated overnight. Forty-eight hours after transfection, the CASR-HEK293A cells were pre-treated for 30 min with physiological saline solution (PSS) (NaCl 125 mM, KCl 4 mM, HEPES 20 mM, D-Glucose 0.1%, NaH_2_PO_4_ 0.8 mM, MgCl_2_ 1 mM, pH 7.45), supplemented with 0.1% BSA. Subsequently, cells were treated in PSS with increasing concentrations of the calcimimetic R568 (Cayman Chemical Company; 50 nM, 100 nM, 0.5 μM, 5 μM, in the presence of 1.5 mM [Ca^2+^]_o_). Treatments were carried out for 10 min, 60 min, or for 6 h to perform analysis of total protein, subcellular protein fractions, or gene expression, respectively. Untreated cells (NT) were used as controls. The experiments were also performed on cells transfected with the empty vector to verify the specificity of R568 effects on the CASR receptor. PAds-derived cells were treated with the CASR agonist R568 in the same experimental setting used for CASR-HEK293A cells for 60 min or 6 h to perform proteins and gene expression analysis, respectively.

### 4.11. Treatment of CASR-HEK293A Cells with Rho-Kinase Inhibitors 

To investigate the potential role of downstream effectors RhoA/ROCK in CASR-mediated nuclear translocation of YAP1, 48 h after transfection, serum-starved CASR-HEK293A cells were pre-treated for 1 h with either 10 μM Y-27632 or 1 μM H-1152 (both from Sigma–Aldrich) in PSS supplemented with 0.1% BSA and 1.5 mM [Ca^2+^]_o_. Subsequently, the cells were stimulated for 1 h with increasing concentrations of R568 (as described above) in the presence or absence of the Rho-kinase inhibitors. Untreated cells (NT) were used as controls.

### 4.12. Statistical Analysis

Gene expression data were log2 transformed and presented as mean±SEM. Protein expression levels were presented as fold change mean±SEM versus not treated (NT) condition. Differences among levels at each experimental time point were tested by one-way ANOVA analysis followed by Bonferroni’s correction post-test. Correlations between gene expression levels were tested by linear regression analysis. A probability value (*p*) less than 0.05 was considered statistically significant. Statistical analyses were performed using Prism v6.0 (GraphPad Inc., San Diego, CA, USA). 

## 5. Conclusions

In conclusion, we investigated the Hippo/YAP1 pathway in parathyroid tumors. YAP1 emerges as a transcriptional factor coupled to CASR activation through ROCK: CASR activation induces YAP1 transcriptional action, and resistance to [Ca^2+^]_o_ may reduce YAP1 nuclear accumulation. Data suggest that YAP1 may act as an oncosuppressor in parathyroid tumor cells. Though the role of YAP1 in parathyroid adenomas needs to be further elucidated, these results expand intracellular signaling aberrations characterizing parathyroid tumors and coupled to CASR, suggesting new targets for medical treatment of parathyroid tumors, which represents an unmet clinical need in the management of both benign and malignant parathyroid tumors.

## Figures and Tables

**Figure 1 ijms-22-02016-f001:**
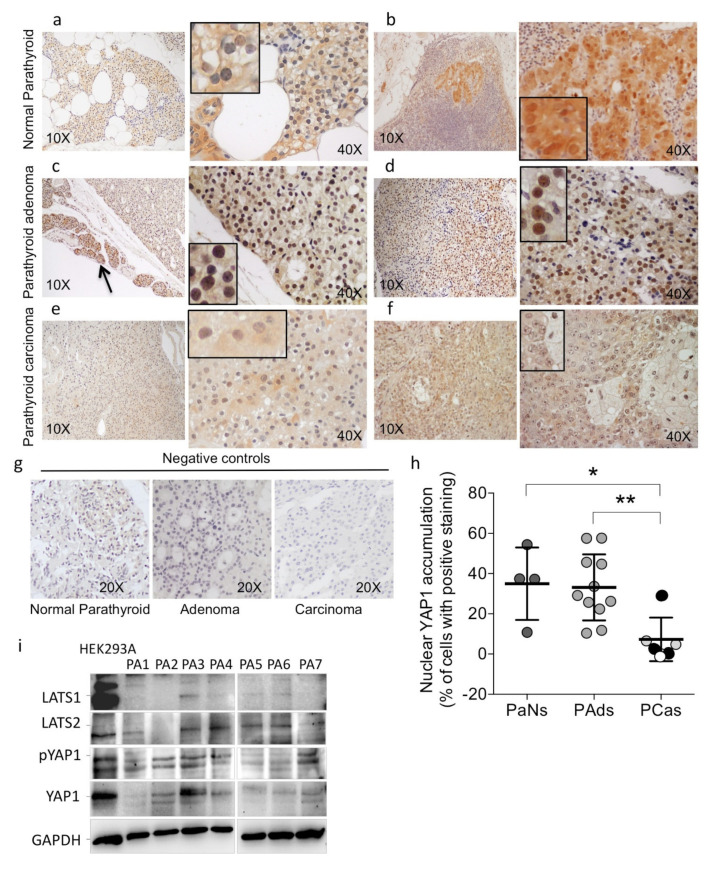
Expression of the Hippo pathway members Yes-associated protein 1 (YAP1) and LATS1/2 (large tumor suppressor 1/2) in parathyroid tissues. Immunohistochemistry analysis for total YAP1. Representative images showed immunostaining in normal parathyroid glands from normocalcemic patients (**a**,**b**), benign parathyroid adenomas (**c**,**d**), parathyroid carcinomas (**e**,**f**). (**c**) Black arrow indicates normal parathyroid cells of the parathyroid gland rim surrounding the parathyroid adenoma. (**e**) Section of parathyroid carcinoma from a patient harboring a germline inactivating *multiple endocrine neoplasia type 1 (MEN1)* gene mutation. Magnifications are indicated in each panel; the inserts show enlarged details. (**g**) Negative controls. (**h**) Quantification of nuclear YAP1 staining as a percentage of positive parathyroid cells; each dot is a case; lines, mean, and SD; PaNs, parathyroid normal glands from normocalcemic patients; PAds, parathyroid benign adenoma; PCas, parathyroid carcinoma; black or grey circles, PCas harboring *cell division cycle 73 (CDC73)* or *MEN1* inactivating mutations, respectively; white circle, PCa harboring *CDC73* and *MEN1* wildtype alleles; *, *p* = 0.024; **, *p* = 0.012 by one-way ANOVA corrected for multiple comparisons. (**i**) Western blot analysis of LATS1, LATS2, phosphorylated YAP1, and total YAP1 expression, in total protein extracts from a series of seven parathyroid adenomas (PA), compared with total protein extracts from human embryonic kidney 293A (HEK293A) cells. GAPDH was a loading control.

**Figure 2 ijms-22-02016-f002:**
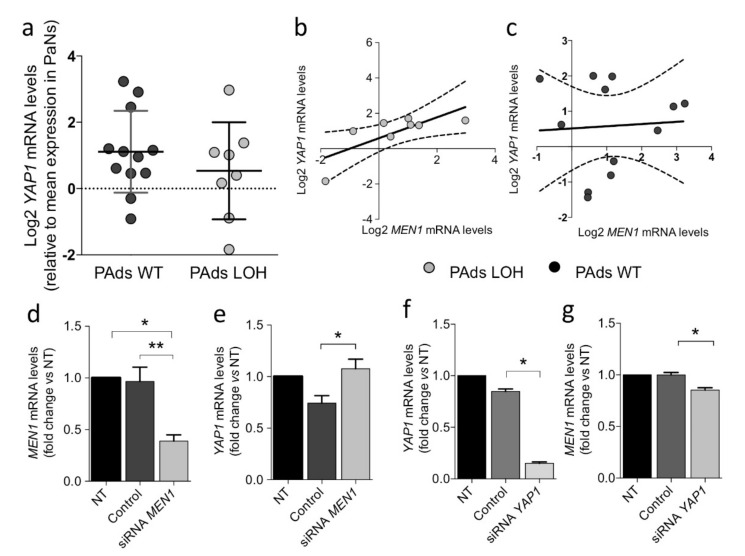
*YAP1* expression and chromosome 11 status in parathyroid adenomas. (**a**) *YAP1* gene expression levels in parathyroid adenomas (Pads) with a normal haplotype for chromosome 11 (PAds WT) or PAds harboring loss of heterozygosity of the chromosome 11 (PAds LOH); *YAP1* mRNA levels were relatively quantified on that detected in three normal parathyroid glands and log2-transformed. (**b**) Correlation between *YAP1* and *MEN1* gene expression levels in PAds harboring the chr.11 LOH (r^2^ = 0.561, *p* = 0.032), and (**c**) in PAds with no loss of chr.11 (WT). (**d**) Transient downregulation of *MEN1* by siRNA in PAds-derived cells (*n* = 4) was verified by qPCR relative to cultures treated with a mock siRNA (Control) or left untreated (NT); * *p* = 0.006; ** *p* = 0.011 by one-way ANOVA. (**e**) *YAP1* mRNA expression was analyzed by qPCR after transient transfection with a *MEN1* or a control siRNA in primary PAd cultures and expressed relative to untreated (NT) samples; * *p* = 0.036 by one-way ANOVA. (**f**) Transient downregulation of the *YAP1* or a control transcript by siRNA in HEK293A cells (*n* = 3) was verified by qPCR relative to untreated samples (NT); *, *p* = 0.008 by one-way ANOVA. (**g**) *MEN1* expression was analyzed by qPCR in HEK293A cultures after transient transfection with a *YAP1* or a control siRNA (*n* = 3) and expressed relative to untreated samples (NT); * *p* = 0.021 by one-way ANOVA.

**Figure 3 ijms-22-02016-f003:**
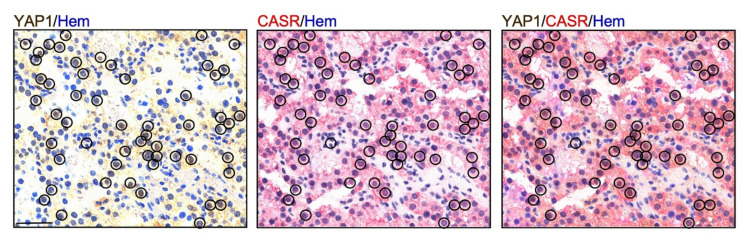
Coexpression of YAP1 and calcium-sensing receptor (CASR) in parathyroid tumor cells. Representative images of the double immunohistochemistry (IHC) are shown for each staining as well as the double staining. Scale bar, 50 μm.

**Figure 4 ijms-22-02016-f004:**
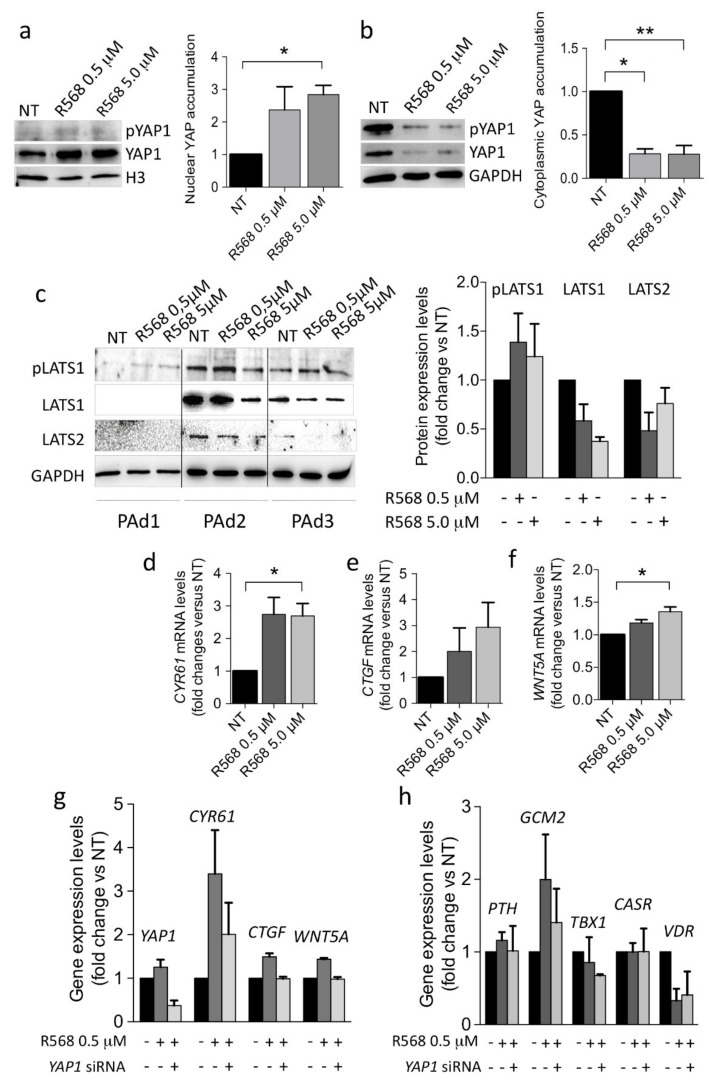
CASR activation induces YAP1 signaling in primary PAd cultures. (**a**,**b**) YAP1 nuclear accumulation (**a**) or cytoplasmic abundance (**b**) were investigated in primary PAd cultures (*n* = 3) after stimulation for 1 h with the indicated concentrations of the CASR agonist R568 (* in panel **a**, *p* = 0.06; * in panel **b**, *p* = 0.018 in presence of 0.5 μM, and ** in panel **b**, *p* = 0.037 in presence of 5.0 μM); representative Western blots are shown with densitometric analyses (right panels). Bars, mean protein expression from three experiments relative to untreated cultures (NT) ± SEM. *p*-values are from one-way ANOVA. (**c**) Inhibition of the LATS1/2 protein levels in primary PAd-cultures (*n* = 3) upon treatment with the CASR agonist R568. Western blot and relative densitometric analysis (right panel) are shown. (**d**−**f**) Analysis by qPCR of the indicated YAP1-target genes expression in three primary PAds-derived cultures treated with R568 (* *p* = 0.037 in panel **d**; * *p* = 0.210 in panel **e**, and * *p* = 0.035 in panel **f**) by qPCR in 3 PAds-derived cell preparations. (**g**,**h**) The expression of the YAP1-targeted genes *CYR61*, *CTGF*, and *WNT5A* or of the key parathyroid genes *PTH*, *GCM2*, *TBX1*, *CASR*, and *VDR* was analyzed in primary PAds-cultures transiently silenced for *YAP1* and treated with the CASR agonist R568. Data are expressed relative to untreated (black bars). Bars, mean gene expression from two experiments ±SEM.

**Figure 5 ijms-22-02016-f005:**
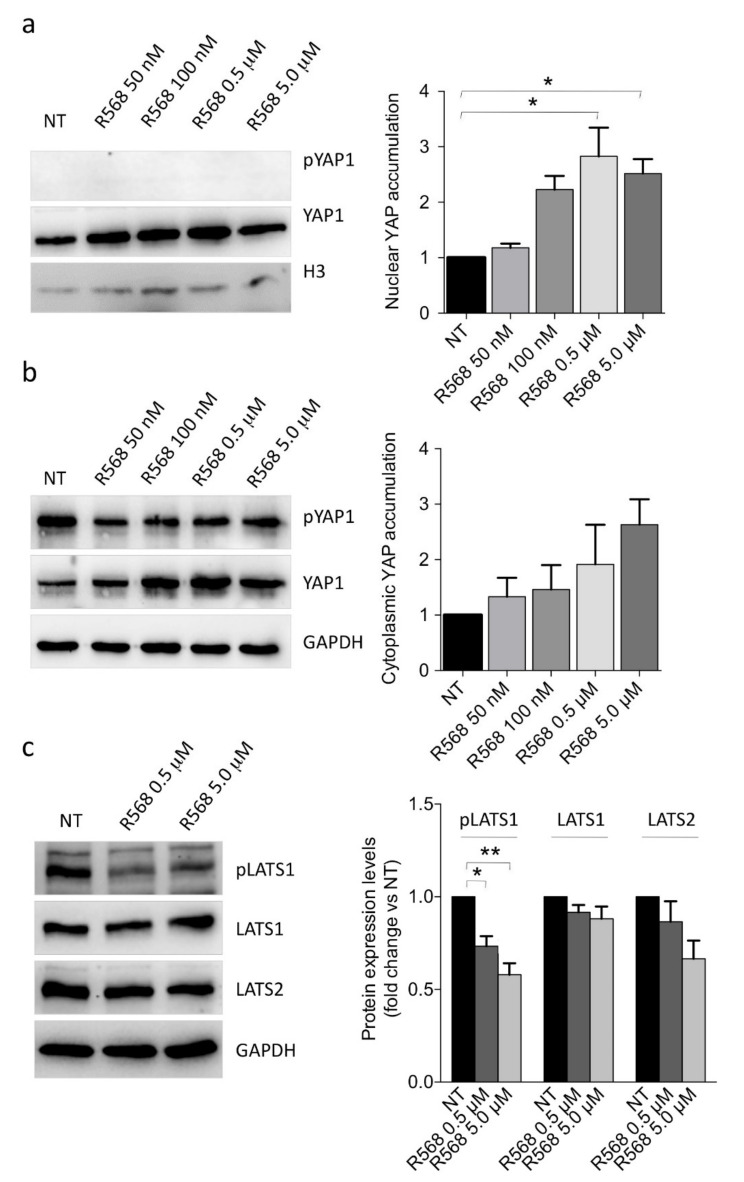
CASR activation induces the YAP1 signaling in the CASR-HEK293A cell model. (**a**,**b**) Accumulation of YAP1 in the nuclear (**a**) or the cytosolic (**b**) fractions of CASR-HEK293A cells stimulated for 1 h with increasing concentrations of the CASR agonist R568 (* *p* = 0.003 by one-way ANOVA) was analyzed by Western blotting, and a representative blot is shown. (**c**) Expression of phosphorylated LATS1 (pLATS1), LATS1, and LATS2 in CASR-HEK23A cells treated with the indicated concentrations of R568 for 10 min; a representative Western blot is shown (* *p* = 0.006 vs. NT; ** *p* = 0.0003 vs. NT, by one-way ANOVA). Bars, mean ± SEM (*n* = 3) of the fold change with respect to untreated conditions (NT).

**Figure 6 ijms-22-02016-f006:**
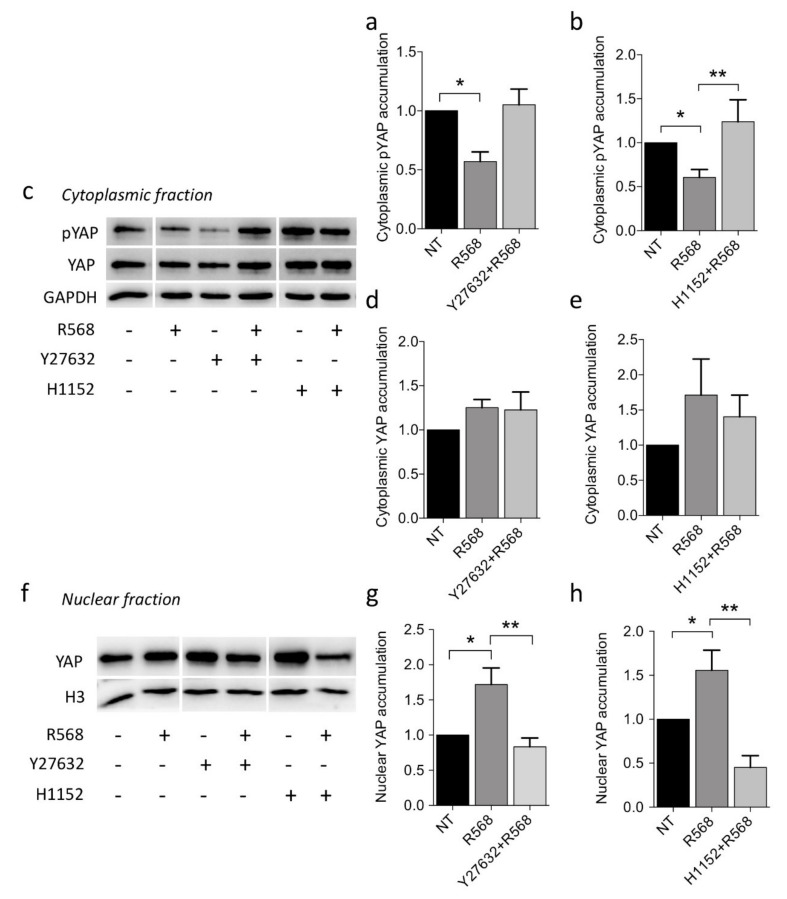
YAP1 modulation is coupled with CASR activation through RhoA/rho-associated protein kinase (ROCK) signaling in CASR-HEK293A cells. (**a**–**e**) Phosphorylated (panels **a**–**c**) or total (panels **c**–**e**) cytoplasmic YAP1 (pYAP1) expression in CASR-HEK293A cells incubated for 1 h with R568 and the ROCK inhibitors Y27632 (panel **a**, 10 μM; * *p* = 0.006) or H1152 (**b**, 1 μM; * *p* = 0.014; ** *p* = 0.083). A representative Western blot is shown in panel **c**. (**f**–**h**) CASR-HEK293A cells were incubated with the ROCK inhibitors Y27632 (panel **h**, 10 μM; *, *p* = 0.057; ** *p* = 0.047) and H1152 (panel **h**, 1 μM; * *p* = 0.051; ** *p* = 0.020 by one-way ANOVA). Then nuclear YAP1 accumulation was analyzed by Western blotting and quantified by densitometric analysis. Bars, mean ± SEM (*n* = 3); *p*-values are from one-way ANOVA.

**Figure 7 ijms-22-02016-f007:**
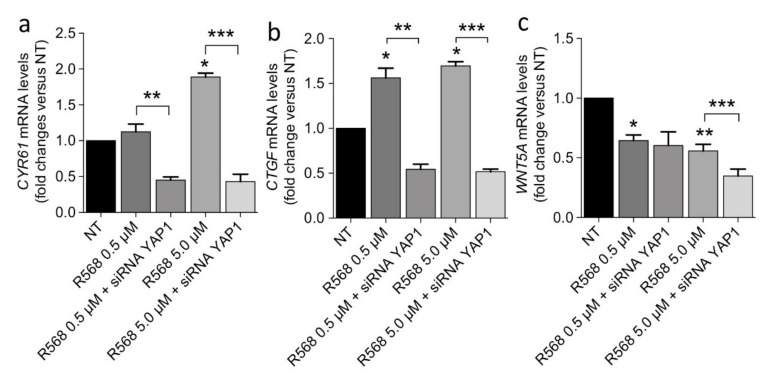
CASR-YAP1 signaling modulation in the CASR-HEK293A cell model. CASR-HEK293A cultures were transiently silenced with a YAP1- or a mock-siRNA and then treated with R568. Gene expression of the indicated transcripts was analyzed by qPCR. *p*-values are from one-way ANOVA and are as follows: CYR61 (panel **a**): * *p* = 0.001 vs. NT; ** *p* = 0.012; *** *p* = 0.0001; CTGF (panel **b**): * *p* = 0.001 vs. NT; ** *p* = 0.031; *** *p* = 0.0001; WNT5A (panel **c**): * *p* = 0.015; ** *p* = 0.0002 vs. NT; ***, *p*= 0.01. Gene expression levels are presented as fold change with respect untreated conditions; data are mean ± SEM (*n* = 4).

**Figure 8 ijms-22-02016-f008:**
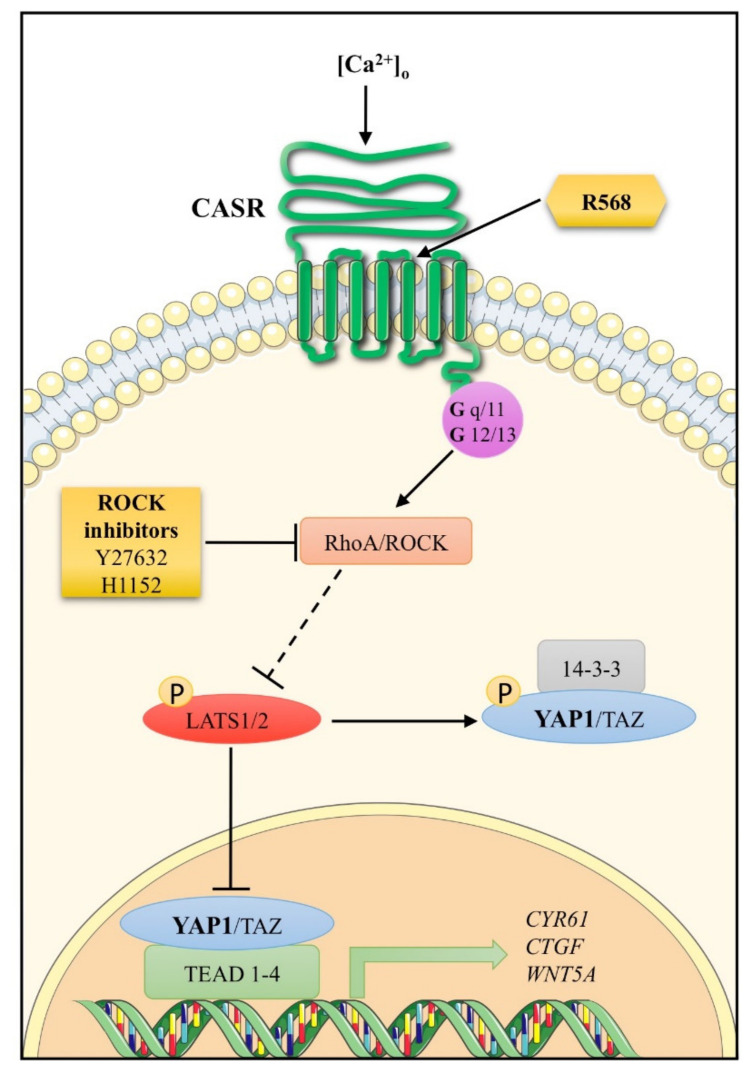
Schematic representation of the CASR-ROCK-YAP1 signaling pathway in tumor parathyroid cells. Stimulation of the calcium-sensing receptor with Ca^2+^ or R568 through the coupling with Gα12/13 or Gαq/11 leads to activation of small GTPase protein RhoA, which indirectly inhibits LATS kinase activity. LATS1/2 phosphorylates YAP1 on highly conserved residues of serine. The phosphorylation of YAP1 promotes cytoplasmic retention and degradation. However, when RhoA inhibits LATS1/2, dephosphorylated YAP1 translocates into the nucleus, where it binds and activates TEAD transcription factors, inducing the expression of multiple genes. Part of this figure was created with images from Servier Medical Art (http://smart.servier.com/, accessed on 1 January 2021), licensed under the Creative Commons Attribution 3.0 Unported License (https://creativecommons.org/licenses/by/3.0/, accessed on 1 January 2021).

**Table 1 ijms-22-02016-t001:** Genetic, biochemical, and clinical features of the series of parathyroid carcinomas (PCas) analyzed by immunohistochemistry for YAP1 expression. *Cell division cycle 73 (CDC73)*, cell division cycle 73; *MEN1*, multiple endocrine neoplasia type 1; SCa, serum total calcium; PTH, plasma parathormone: Recur., tumor local recurrence; Metas., distant metastasis; WT, wildtype gene sequence. All samples were included in previous studies [25,47,48,49].

PCa Sample	Germline/ Somatic	*CDC73* Mutation	*MEN1* Mutation	SCa mg/dL	PTH pg/mL	Recur.	Metas.
PCa1	Germline	c.518_521del4	WT	13.0	660	yes	no
PCa2	Germline	p.R415X	WT	12.7	367	no	yes
PCa3	Germline/Somatic	WT	WT	20.2	940	no	no
PCa4	Germline	WT	p.D418N	Na	na	no	no
PCa5	Germline	c.685_688 delAGAG	WT	10.0	101	yes	no
PCa6	Germline	WT	p.Q166L/A167S	11.8	160	yes	no

**Table 2 ijms-22-02016-t002:** Specific probes used for gene expression analysis.

Gene Name	Probe Number
*YAP1*	Hs00371735_m1
*GCM2*	Hs00899403_m1
*PTH*	Hs00757710_g1
*TBX1*	Hs00271949_m1
*YAP1*	Hs00371735_m1
*CYR61*	Hs00155479_m1
*CTGF*	Hs00170014_m1
*CASR*	Hs01047795_m1
*WNT5A*	Hs00998537_m1
*VDR*	Hs01045843_m1
*MEN1*	Hs00365720_m1
*HMBS*	Hs00609297_m1
*B2M*	Hs99999907_m1

## Data Availability

The datasets generated and analyzed during the current study are available at https://doi.org/10.5281/zenodo.4525509.

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
