# Peer review of "Yes-Associated Protein 1 Is a Novel Calcium Sensing Receptor Target in Human Parathyroid Tumors"

_ijms, 2021, doi:10.3390/ijms22042016_

Round 1

Reviewer 1 Report

In the current study, the authors investigated the Hippo/YAP1 pathway in parathyroid tumors and relation to CASR activation, in an objective and analytical manner. The topic is relatively novel in the field of parathyroid tumors, as previous studies had already recognized the pivotal role of Hippo pathway in organ size control, tissue regeneration, and tumorigenesis in different types of cancer, but it has been not yet studied in parathyroid.

The study provides an extensive amount of interesting data, but they should be introduced with more precision and accuracy, as reading may be difficult at some text passages and an attempt to synthetize some findings should be worthwhile. 

The purpose of the study is detailed in the manuscript, but authors should better highlight the aim in the title and abstract to focus the interest of the reader.

A more appropriate title could be chosen, since the authors announce "identification and characterization of CASR" in their title but as stated in lines 103-110 and in the conclusions, they offer a more detailed analysis of the Hippo/YAP1 pathway.

Similarly, the abstract could be reviewed, as it does not clearly summarize background and findings.

In introduction there are some points to be clarified. In explanation of Hippo pathway a figure could be helpful. The physiological role of other genes and protein complex studied (LATS2 and ROCK) should be explained (line 99).

The captions under the figures are well detailed, so the explanations in the text could be synthetized without compromising the understanding.

Line 88: calcium sensing receptor, use acronym.

Author Response

Authors’ responses to Editor and Reviewers’ comments

Manuscript #: IJMS-1069959

We thank the Editor and Reviewers for considering our manuscript and for providing insightful comments. Accordingly, we addressed your concerns and comments. Please find below our point-by-point responses (in blue); modifications in the text are tracked for presentation clarity.

Reviewer 1 comments:

  1. Data should be introduced with more precision and accuracy. Better highlight the aim in the title and abstract to focus the interest of the reader.

We thank the Reviewer for his/her suggestions. Accordingly, we changed the manuscript title into “Yes-associated protein 1 (YAP1) is a novel calcium-sensing receptor target in human parathyroid tumors” and we rewrote the abstract.

  1. Introduction: In explanation of Hippo pathway a figure could be helpful. The physiological role of other genes and protein complex studied (LATS2 and ROCK) should be explained (line 99).

Following the Reviewer suggestion, we described the role of the different Hippo pathway components in the Introduction, page 2 lines 65-77 and page 3 lines 102-108. A schematic of pathway is provided in the Discussion section (Figure 8).

  1. Explanations of results in the text could be synthetized.

Results have been revised and shortened.

  1. Line 88: calcium sensing receptor, use acronym.

CASR has been used.

Reviewer 2 Report

The authors reported differential YAP1 expression in parathyroid tumors compared to normal parathyroid tissues and observed YAP1 accumulation downstream to CaSR activation in CaSR-HEK293A cells. The main issue of this report is that CaSR expression is generally reduced in parathyroid tumors. Thus, clinical observation and experimental findings are inconsistent.

(1) Although the authors reported that nuclear YAP1 expression was similar between PaNs and PAds, representative photographs shown in Figure 1 suggest that nuclear YAP1 expression appears increased in adenomas. More clinical samples are required to validate this finding.

(2) A few studies (two examples shown below) have shown that CaSR expression is decreased in parathyroid tumors. The authors should clarify this issue by analyzing the association between the expressions of CaSR and YAP1 in clinical samples.
Endocr Pathol 2018;29:250-8. doi: 10.1007/s12022-018-9524-9.
J Clin Endocrinol Metab 2020;105:3015-24. doi: 10.1210/clinem/dgaa419.

(3) An alternative explanation is that although CaSR activation could induce YAP1 accumulation and upregulation of target genes (CYR61, CTGF, and WNT5A), the differential YAP1 expression in clinical parathyroid tumors is actually downstream of other factors.

(4) The biologic role of MEN1 (Figure 2) remains unclear from this report.

Author Response

Authors’ responses to Editor and Reviewers’ comments

Manuscript #: IJMS-1069959

We thank the Editor and Reviewers for considering our manuscript and for providing insightful comments. Accordingly, we addressed your concerns and comments. Please find below our point-by-point responses (in blue); modifications in the text are tracked for presentation clarity.

Reviewer 2 comments:

  1. The main issue of this report is that CaSR expression is generally reduced in parathyroid tumors. Thus, clinical observation and experimental findings are inconsistent.

We acknowledge the point rose by this Reviewer, nevertheless we want to highlight that CASR is a key molecule both in parathyroid physiology and pathophysiology. Further, patients with parathyroid tumor often suffer of hypercalcemia. Therefore, we think that a deeper understanding of CASR responsive genes and/or pathway members is relevant for getting new insights into parathyroid tumorigenesis and possible novel therapeutic approaches. Here we show that CASR and YAP1 are interconnected in parathyroid cells and in a cell model of CASR activity (HEK293A cells overexpressing the CASR). Therefore, we think that our findings, although preliminary, are relevant for the understanding of parathyroid tumor pathobiology. Limitations of our study have been discussed at page 12 lines 361-364.

  1. Nuclear YAP1 expression in the different tissue types.

To corroborate our findings we digitally scanned parathyroid tissues stained for YAP1 using Aperio instrument and counted positive nuclei using a nuclear algorithm within the ImageScope software. Quantifications are now presented in the new panel h of Figure 1. Accordingly, also the paragraph 2.1 of the Results section has been rewritten.

  1. A few studies (two examples shown below) have shown that CaSR expression is decreased in parathyroid tumors. The authors should clarify this issue by analyzing the association between the expressions of CaSR and YAP1 in clinical samples.
    Endocr Pathol 2018;29:250-8. doi: 10.1007/s12022-018-9524-9.
    J Clin Endocrinol Metab 2020;105:3015-24. doi: 10.1210/clinem/dgaa419.

We thank the Reviewer for making this point. Accordingly we performed a double immunohistochemistry for YAP1 and CASR in a subset (n=6) of parathyroid adenomas. Results show that YAP1-positive cells also express CASR at their plasma membrane. This is now shown in a new Figure 3 and presented at page 6, lines 197-200. Furthermore, the suggested references have been added in the text.

  1. An alternative explanation is that although CaSR activation could induce YAP1 accumulation and upregulation of target genes (CYR61, CTGF, and WNT5A), the differential YAP1 expression in clinical parathyroid tumors is actually downstream of other factors.

We agree with the Reviewer’s comment and we acknowledge this point at page 12 lines 362-363 of the Discussion.

  1. The biologic role of MEN1 (Figure 2) remains unclear from this report. We agree with the Reviewer. Therefore, discussion of MEN1 role in CASR-YAP1 axis has been revised at page 12 lines 361-362.

Round 2

Reviewer 2 Report

The authors have addressed all my concerns.

Author Response

My co-Authors and I sincerely thank the Reviewer for the positive comment about the manuscript.

English language has been carefully revised.